# Long-term mutual phase locking of picosecond pulse pairs generated by a semiconductor nanowire laser

B. Mayer[1,2], A. Regler[1,3], S. Sterzl[1], T. Stettner[1], G. Koblmüller[1], M. Kaniber[1], B. Lingnau[4], K. Lüdge[4] & J.J. Finley[1]

The ability to generate phase-stabilized trains of ultrafast laser pulses by mode-locking underpins photonics research in fields, such as precision metrology and spectroscopy. However, the complexity of conventional mode-locked laser systems has hindered their realization at the nanoscale. Here we demonstrate that GaAs-AlGaAs nanowire lasers are capable of emitting pairs of phase-locked picosecond laser pulses with a repetition frequency up to 200 GHz when subject to incoherent pulsed optical excitation. By probing the two-pulse interference spectra, we show that pulse pairs remain mutually coherent over timescales extending to 30 ps, much longer than the emitted laser pulse duration ($\leq 3$ ps). Simulations performed by solving the optical Bloch equations produce good quantitative agreement with experiments, revealing how the phase information is stored in the gain medium close to transparency. Our results open the way to phase locking of nanowires integrated onto photonic circuits, optical injection locking and applications, such as on-chip Ramsey comb spectroscopy.

[1] Walter Schottky Institut and Physik Department, Technische Universität München, Am Coulombwall 4, Garching 85748, Germany. [2] IBM Research—Zürich, Säumerstrasse 4, Rüschlikon 8803, Switzerland. [3] Institute for Advanced Study, Technische Universität München, Lichtenbergstrasse 2a, Garching 85748, Germany. [4] Institute of Theoretical Physics, Technische Universität Berlin, Hardenbergstraße 36, Berlin 10623, Germany. Correspondence and requests for materials should be addressed to B.M. (email: benedikt.mayer@wsi.tum.de) or to J.J.F. (email: finley@wsi.tum.de).

Wavelength-scale coherent optical sources are vital for a wide range of applications in nanophotonics ranging from metrology[1] and sensing[2] to nonlinear frequency generation[3] and optical switching[4]. In these respects, semiconductor nanowires (NWs) are of particular interest since they represent the ultimate limit of downscaling for photonic lasers with dielectric resonators[5]. By virtue of their unique one-dimensional geometry, NW lasers combine ultra-high modal gain, support low-loss guided modes and facilitate low threshold lasing tunable across the ultraviolet, visible and near infrared spectral regions[6–9]. Recently, optically pumped NW lasers have been demonstrated at room temperature and they can now be site-selectively integrated onto silicon substrates[10,11]. While the fundamental carrier relaxation and gain dynamics of NW lasers have been explored[6,12], the coherent dynamics have hitherto received comparatively little attention. Here we demonstrate that subsequently emitted ultrafast ($\leq 3$ ps duration) laser pulses emitted from incoherently pumped GaAs-AlGaAs core-shell NW lasers remain mutually phase coherent over timescales that are approximately 10 times longer than the emitted pulse duration. The mechanism responsible for the mutual phase locking is shown to be linked to coherent dynamics that occur in the postlasing regime when the system remains close to transparency over timescales approaching the spontaneous emission lifetime ($\sim 0.6$ ns). Numerical simulations performed by solving the optical Bloch equations produce good quantitative agreement with experiments supporting our identification of the mechanism that establishes mutual phase coherence.

## Results

**Time-resolved pump-probe response.** Figure 1b shows typical spectral data recorded as a function of the time delay between pump and probe ($\Delta t = t_{pump} - t_{probe}$) for $P_{pump}$ close to $\sim 3 P_{th}$ and three different values of $P_{probe}$: far below threshold (labelled L-SE), close to threshold (L-ASE), and above threshold (L-L). Note that positive $\Delta t$ means that the pump pulse arrives first at the NW. Pronounced interference fringes are observed in the time-integrated spectra recorded for all three combinations of $P_{pump}$ and $P_{probe}$ and persist even for $\Delta t \geq 40$ ps. To check that the observed fringes do indeed arise from the interference between two successively emitted laser pulses, the fringe separation in the frequency domain $\Delta f$ is plotted in Fig. 2 as a function of $\Delta t$. As can be seen, $\Delta f$ decreases inversely with $|\Delta t|$ as expected, indicative of the interference between subsequently emitted NW laser pulses generated by the pump and probe pulses[13]. The data indicate that extremely high maximum repetition rates $\Delta f > 200$ GHz are possible, corresponding to emitted pulse durations $t_{pulse} < 3$ ps. Two-pulse interference was recently reported for GaN, CdS and ZnO NW lasers in operation regimes corresponding to the direct temporal overlap between the emitted laser pulses[6,12]. In contrast, in Fig. 1 we note that interference is still observed in the time-integrated spectrum over timescales more than one order of magnitude longer than the duration of the emitted laser pulses themselves. This surprising observation clearly indicates that coherence is preserved in the NW laser over long timescales after lasing has ceased.

Our experimental observations are found to be in good agreement with the predictions of a numerical model of the pulsed, driven laser system obtained in the framework of the well-known semiconductor Bloch equations for lasers[14,15] with microscopically motivated carrier dynamics[16] (see Methods section). A stochastic spontaneous emission noise source was introduced to simulate the effects of dephasing[17,18]. The Bloch equations are extended by an additional equation describing the time-dependent carrier occupation in the reservoir, defined by the

electronic states addressed by the excitation pulses tuned to an energy of 1.59 eV, and the resulting incoherent scattering processes into the lasing level at 1.51 eV[16]. Figure 1c shows results of our numerical simulations. The best quantitative agreement with our experimental observations was obtained for a photon lifetime in the resonator of $\tau_{ph} \sim 1$ ps, obtained from the measured Q-factor of the resonator modes, carrier lifetimes in the reservoir of $T_2 \sim 10$ ps and a spontaneous emission lifetime of the lasing state equal to $T_1 \sim 0.6$ ns. We note that the experimental data exhibits weaker emission for $|\Delta t| \leq 8$ ps as compared to the numerical modelling presented in Fig. 1. As discussed in Supplementary Note 1, this arises from saturation of the excitation reservoir due to Pauli blocking when the probe pulse is incident on the sample before lasing due to the pump pulse has been fully established (Supplementary Fig. 1)[12]. In our experiment, such reservoir saturation effects are expected for excitation levels exceeding approximately $4 P_{th}$, with $P_{th}$ being the threshold pump power, as can clearly be seen in the final three data points of the input–output curve presented in Supplementary Fig. 1. In our modelling, we chose not to implement these reservoir saturation effects, focussing instead on the theoretical description of phase-retention over long timescales. However, including such effects in the modelling can indeed reproduce the weaker emission close to $\Delta t = 0$ in Fig. 1 (Supplementary Fig. 2). The simulation reproduces most of the principle features observed in the experiment, including the two-pulse interference and the time-dependent redshift caused by a small detuning between lasing transition and cavity mode, measured to be 1.7 meV (Supplementary Fig. 1). This good qualitative agreement in form between experiment and simulations confirm the interpretation of the observed features presented in Fig. 1b) as being caused by two-pulse interference in the time-integrated spectral response of the NW lasers studied.

**Time-domain analysis of coherent pulse pairs.** We continue to explore the pump–probe dynamics of the NW lasers in the time domain by computing the discrete Fourier transform of the experimental spectra presented in Fig. 1 to obtain the temporal evolution of the emitted intensity $\Delta \tau$ as a function of the separation between the pump and probe pulses, $\Delta t$. A similar approach has recently been used in ref. 11 to analyse two-pulse spectral interference in the emission from ZnO NWs, albeit over shorter timescales when the two emitted laser pulses temporally overlap in the detection system. The results of this procedure are presented in Fig. 3 together with the corresponding numerical simulations. The figure compares data recorded for a situation with $P_{pump}$ in the lasing regime, that is, pump intensities $P$ larger than the threshold pump energy $P_{th}$ and two different probe intensities: $P_{probe} \ll P_{th}$ in the spontaneous emission regime, labelled by L-SE (Fig. 3a), and a situation where both $P_{pump}$ and $P_{probe}$ are in the lasing regime labelled L-L (Fig. 3b). The colour plots reveal that two temporally distinct NW laser pulses are generated at times $\Delta \tau = 0$ (pulse-1) and $\Delta \tau > 0$ (pulse-2), provided that $P_{pump}$ is above threshold (for example, $P_{pump}/P_{th} > 2$) and the probe pulse arrives after the pump to enhance the gain in the NW. Note that for negative detuning ($\Delta t < 0$), when the weaker probe pulse arrives before the stronger pump, no second pulse is seen as expected (labelled SE-L in Fig. 3a) corresponding to an absence of interference between the emission generated by pump and probe beams. In the L-SE configuration, the interference observed in the time-integrated spectrum is indicative that the residual gain in the NW is sufficient for the weak probe pulse to re-establish lasing and that the two pulses are mutually phase coherent despite having no

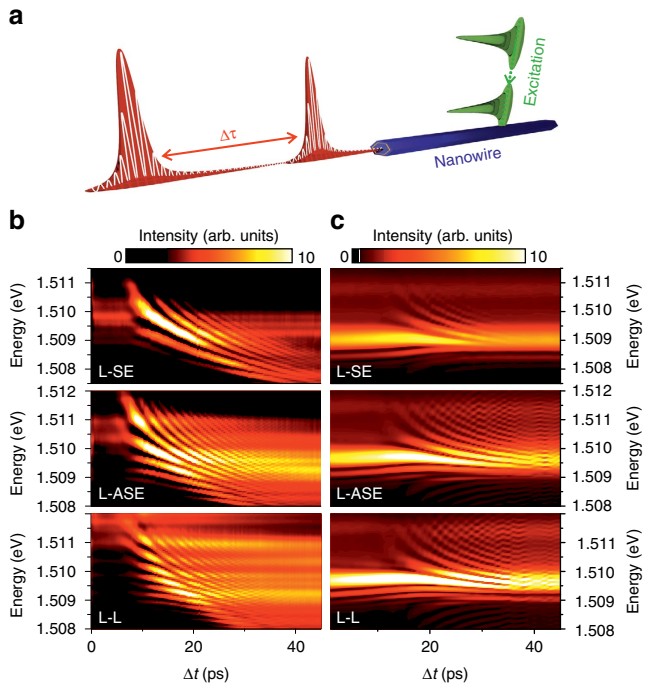

**Figure 1 | Time-resolved pump–probe response of the NW laser.**
(**a**) Schematic illustration of a NW laser emitting a coherent pulse pair.
(**b**) Optical spectra recorded as a function of the pump–probe delay $\Delta t$.
(**c**) Corresponding theoretical results obtained using the optical Bloch
equation model described in the text. **b,c** depict situations for a fixed pump
pulse power in the lasing regime and three different probe pulse powers:
L-SE: probe pulse in the spontaneous emission regime (probe pulse power
$P_{probe} \sim 0.6 \times P_{th}$), L-ASE: probe pulse in the amplified spontaneous
emission regime ($P_{probe} \sim 1.9 \times P_{th}$) and L-L: probe pulse in the lasing regime
($P_{probe} \sim 4.7 \times P_{th}$). The data are plotted on a linear colour scale.

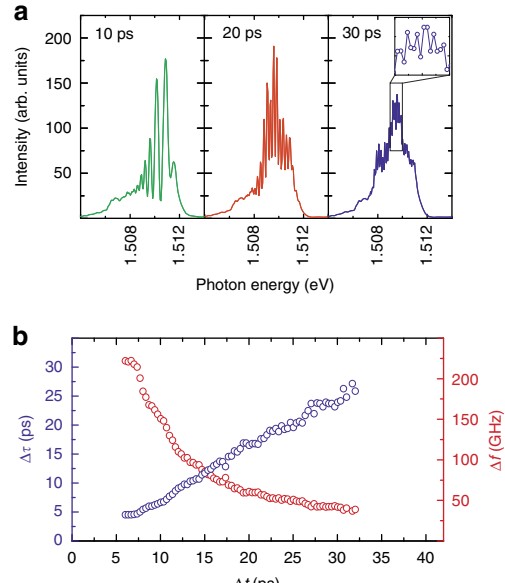

**Figure 2 | Fringe spacing versus pump–probe delay.** (**a**) Selected optical
spectra recorded from the NW laser subject to a pump pulse in the lasing
regime and a probe pulse in the ASE regime (L-ASE) for $\Delta t = +10$ ps
(green), $\Delta t = +20$ ps (red) and $\Delta t = +30$ ps (blue). The inset shows a
zoom-in of the spectrum, demonstrating that modulation is still observed
and that the spacing is close to the resolution limit. (**b**) Repetition rate (red
circles) and NW laser pulse separation (blue circles) as a function of $\Delta t$,
measured from the separation of the interference fringes in the NW laser
spectra and their inverse in **a**, respectively.

temporal overlap at the detector. When both excitation pulses are
in the lasing regime (L-L—Fig. 3b) a clear symmetry is observed
around the point of coincidence ($\Delta t = 0$ ps) as two pronounced
coherent NW laser pulses well separated by a delay $> 30$ ps are
observed for both positive and negative $\Delta t$ (labelled pulse-1 and
pulse-2 in Fig. 3b). The maximum $\Delta \tau$ accessible in our experi-
ments is defined by the limited spectral resolution of the spectral
data presented in Figs 1 and 2. Here the inset at $\Delta t \sim +30$ ps
shows the measured interference fringes sampled at the spectral
resolution $\Delta E = 0.12$ meV that corresponds to $\Delta f = 2.9 \times 10^{10}$ Hz
and an upper temporal resolution $\Delta t = 1/\Delta f \sim 30$ ps. We note that
coming from the experimental data measured in the frequency
domain, a discrete Fourier transform is needed to map the
data into the time domain. As such, explicit information about
the temporal ordering of the two emitted pulses is lost. However,
by comparing the form of the data in Fig. 3 with direct simula-
tions of the same situation (Supplementary Note 2), we assign the
features labelled pulse-1 and pulse-2 to the emission induced by
pump and probe pulses, respectively (Supplementary Fig. 3).
The duration of the first NW laser pulse emitted, represented by
the white line at the bottom of Fig. 3b, is measured to be
$t_{pulse} = 0.9 \pm 0.1$ ps, whereas the second pulse induced by the
probe pulse has a duration ranging from $t_{pulse} = 3.7$ ps
($\Delta t = +15$ ps) to 1.6 ps ($\Delta t = +30$ ps). Therefore, the analysis
in the time domain is in good agreement with the results
obtained from the analysis of the two-pulse interference in the
frequency domain where similar pulse durations of
$t_{pulse} < 5$ ps, corresponding to a 200 GHz fringe separation, were
observed.

**Mutual phase locking of NW laser pulses**. We continue to
explore the origin of the mutual phase locking of the two emitted
laser pulses by considering the dynamics of the coupled electron–
photon system subject to excitation by multiple pulses. In prin-
ciple, the coherence present in the coupled electron–photon
system is preserved if the time-averaged photon density generated
in the cavity field by stimulated processes is $n_{stim} \gg 1$ and if it far
exceeds that generated by spontaneous processes ($n_{spon}$). When
the NW laser is operating far above threshold, the gain is large
and photons are rapidly generated in the cavity mode via sti-
mulated emission. This enhances the coherence of the cavity field
and the contribution from spontaneous processes is negligible by
comparison ($n_{stim}/n_{spon} \gg 1$). Thus the cavity field interacts
coherently with the gain medium leading to Rabi oscillations. As
the excitation dies away and the gain becomes insufficient for self-
sustaining lasing, both stimulated and spontaneous processes add
photons to the cavity field simultaneously and the cavity field
progressively loses coherence due to the phase fluctuations that
eventually dominate when $n_{stim} \approx n_{spon}$. This corresponds to a
postlasing regime where Rabi oscillations are rapidly damped as
the coherence in the coupled light–matter system becomes lost.
To make these arguments quantitative, we compared our
experiments with numerical simulations of the dynamics of the
coupled electronic–photonic system in the NW laser investigated
when subject to pulsed optical excitation. Here we emphasize that
phase coherence is not transferred from the excitation source to
the NW laser output, since the pump pulse at 1.59 eV is absorbed
into the reservoir before phonon-mediated relaxation and carrier
thermalization to the lasing state at 1.51 eV takes place. Carrier
relaxation and thermalization in bulk GaAs typically occurs over
timescales of a few picoseconds[19–21] and our simulations were
performed using an incoherent relaxation time into the lasing
state of $T_2 \sim 10$ ps. The key timescales describing the dynamics of

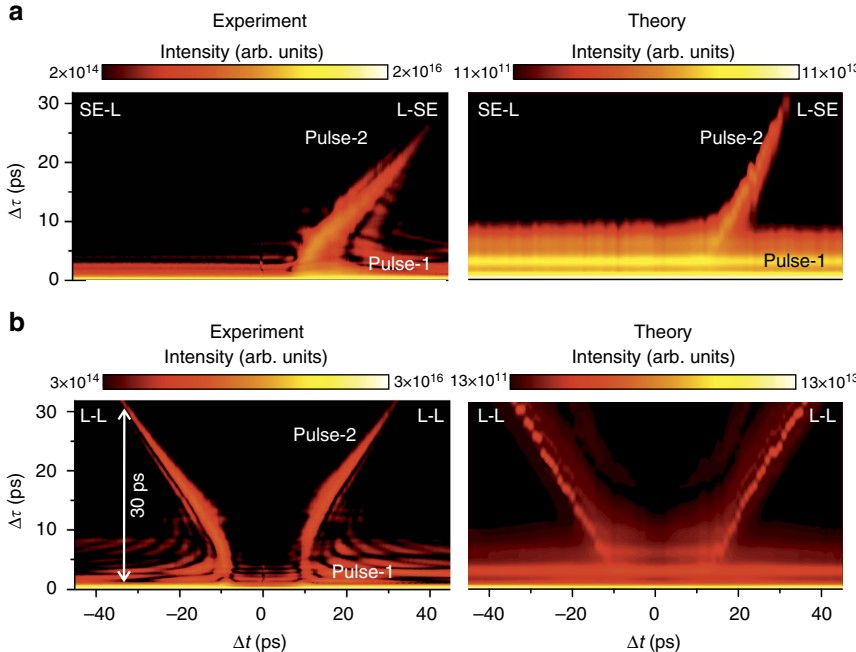

**Figure 3 | Experimental and theoretical pump–probe data in the time domain.** Time-dependent emission of the NW laser as a function of $\Delta t$ after Fourier transforming the optical spectra. The data are plotted on a logarithmic colour scale that spans two orders of magnitude. (**a**) Pump pulse in the lasing regime and probe pulse in the spontaneous emission regime (pump pulse power $P_{pump} \sim 5 \times P_{th}$ and probe pulse power $P_{probe} \sim 0.6 \times P_{th}$). (**b**) Pump and probe pulse in the lasing regime (L-L)—($P_{pump} \sim 5 \times P_{th}$ and $P_{probe} \sim 4.7 \times P_{th}$).

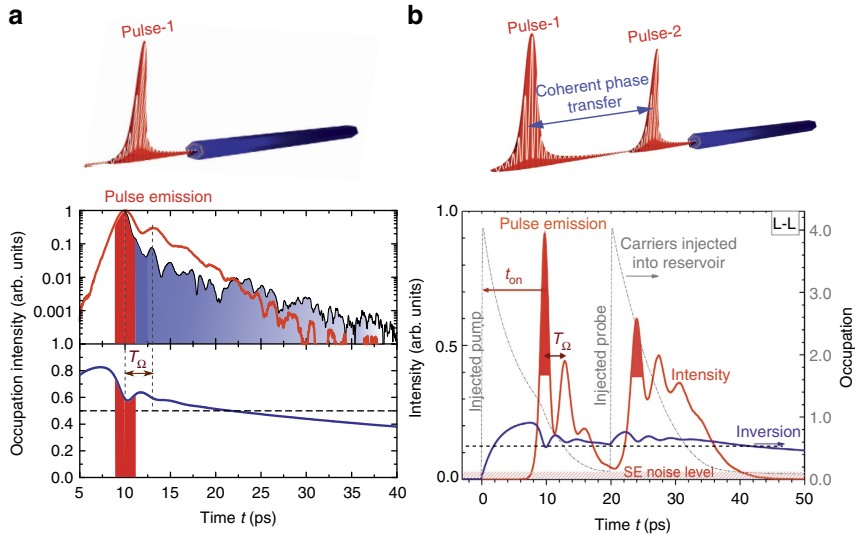

**Figure 4 | Coherent-phase information transfer between two subsequent NW laser pulses.** (**a**) Time dependence of the emitted intensity obtained by Fourier transforming the optical spectra of the NW laser after only the pump pulse with $P/P_{th} = 4$ was injected (red line: simulations, black line: experimental data) and computed occupation of the lasing state (blue line) as a function of time. The schematic illustration depicts a NW laser that emits a single pulse. (**b**) Directly simulated time-dependent emission (red line) of the NW after a pump and probe pulse in the lasing regime (L-L) separated by $\Delta t = 20$ ps were injected (thin dotted line). Blue line shows the inversion of the laser level and the hatched region depicts the spontaneous emission noise level. The red regions in **a,b** mark the short pulse emission during the first Rabi oscillation ($T_\Omega$ marks the Rabi oscillation period and $t_{on}$ the time needed to reach inversion after the pump pulse arrived). The schematic illustration depicts a NW laser that emits a coherent pulse pair.

the coupled electron–photon system are the photon lifetime in the cavity $\tau_{ph} = (2\kappa)^{-1} \sim 1\,\text{ps}^8$, and the polarization lifetime of the lasing state is taken to be $T_p = 5$ ps, typical for bulk GaAs[22,23]. The lasing level is depleted by spontaneous emission with a time constant of $T_1 \sim 0.6$ ns and a pure dephasing time of $R^{-1} \sim 2.5$ ps was used, again both typical values for GaAs at low temperatures[22,23].

Selected numerical results of our modelling are presented in Fig. 4. Figure 4a shows the calculated time dependence of the photon density in the resonator (red curve) and inversion of the lasing state (blue curve) when the NW laser is excited by a single pulse with $P/P_{th} = 4$. The computed NW laser output (red curve) is compared with our experiments (black curve) by Fourier transforming the single-pulse emission spectrum. The time over

which the system remains above threshold, and self-sustaining lasing oscillations occur, is depicted by the red-shaded region on the figure. As shown by the blue curve in Fig. 4a, the inversion in the electronic system remains very close to transparency (occupation close to 0.5) for timescales up to approximately 40 ps, and thereby, a significant fraction of the light generated in the cavity mode is due to stimulated emission, even though self-sustaining lasing is no longer possible since modal gain does not compensate losses. If the second probe pulse is incident on the non-lasing NW at a point in time where both stimulated and spontaneous processes add photons to the cavity field, then the second pulse re-inverts the population and the phase of the lasing oscillation is defined by the weak residual coherence in the coupled electronic–photonic system. This qualitative picture is confirmed by the simulations presented in Fig. 4b that show results obtained when the NW laser is sequentially pumped by pump and probe pulses in the lasing regime (L-L) separated by $\Delta t \sim +20$ ps (grey curves). Again, the blue curve shows the electronic inversion of the laser level, the red curve depicts the photon density in the resonator and the hatched region shows the spontaneous emission (noise) level. Lasing starts at a time $t_{on}$ after the first excitation pulse, after the photon density in the resonator has built up and Rabi oscillations occur before the second pulse again enhances the excitation level. Mutual coherence between the two emitted pulses exists only if the density of coherent photons driving the Rabi oscillations far exceeds the density of spontaneously emitted photons generating background noise that destroy coherence.

Finally, we briefly consider the ultrafast dynamics of the NW laser immediately after pumping. An experimental value for the turn-on time of the NW laser ($t_{on}$) can be found by measuring the peak amplitude of pulse-2 in Fig. 3 in the $\Delta \tau - \Delta t$ plane and extrapolating back to $\Delta \tau = 0$. Whenever the probe pulse power lies firmly in the lasing regime, we observe a linear variation of the maximum amplitude of pulse-2 as expected and estimate that $t_{on} \sim 2$–4 ps for the NW laser investigated. Here we note that the weak periodic oscillation seen in the L-L data presented in Fig. 3b for positive $\Delta t$ (probe follows pump) reflects the onset of Rabi oscillations that occur after emission of the first laser pulse, but the fine structure oscillations for fixed $\Delta \tau$ are an artefact of Fourier transforming the purely real data set. This is confirmed by examining the SE-L ($\Delta t < 0$) region of Fig. 3a, where similar oscillations are observed along the $\Delta \tau$ axis for fixed $\Delta t$, but with vanishing fine structure since the weak probe pulse that excites the NW first does not invert the system and only the later probe pulse induces lasing.

## Discussion

Our results demonstrate how coherent-phase information can be transferred between two subsequently emitted NW laser pulses by virtue of coherent Rabi oscillations that occur during the postlasing transparency regime. The consequent observation of interference fringes reveals the potential for high repetition rates $> 200$ GHz while preserving mutual coherence between the emitted laser pulses. The discrete Fourier transform of the spectra obtained from ultrafast pump–probe experiments exhibits coherent information storage up to timescales of approximately 30 ps and for emitted pulse durations below 3 ps. The ultrafast pulse emission of the NW lasers is followed by strongly damped Rabi oscillations that result from the coherent light–matter interactions in the NW cavity. The experimental results are in good quantitative agreement with theoretical predictions from a microscopically motivated semiconductor Bloch equation model with a realistic noise source that calculates the phase of the photon field and the polarization of the lasing state population in

the NW. The theoretical model reproduces the coherent-phase information transfer between two subsequent NW laser pulses and the generation of the mutually coherent pulses from incoherent excitation. The results present a novel mechanism for transfer of optical phase between NWs integrated into nanophotonic circuits.

## Methods

**Sample preparation.** The GaAs-AlGaAs core-shell NWs were grown using solid source MBE on Si(111) substrates. The substrate was masked by an approximately 2 nm-thick $SiO_2$ layer containing pinholes that act as nucleation sites for the NW growth. The GaAs NWs were synthesized at 610 °C in a self-catalysed (Ga-droplet mediated) vapour–liquid–solid growth mode using Ga and As fluxes of 0.025 and 0.103 nm s$^{-1}$, respectively. In order to provide waveguiding inside the NW gain medium, a subsequent overgrowth step at reduced temperatures of 490 °C and increased arsenic and gallium fluxes of 0.19 and 0.17 nm s$^{-1}$, respectively, was used to increase the NW diameter to approximately 320 nm. To strongly enhance the optical efficiency of the NWs, the GaAs NW core was overgrown with a thin AlGaAs passivation layer using an Al flux of 0.057 nm s$^{-1}$. The growth was finalized by growing a $5 \pm 1$-nm-thick GaAs protection layer on the NW surface[8]. After growth, individual NWs were mechanically removed from the growth substrate and dispersed onto glass whereupon the lasing behaviour of individual NWs could be explored when subject to optical pumping.

**Optical characterization.** All measurements reported in this paper were performed with the sample held at a lattice temperature of $T \sim 20$ K. In experiments performed using single excitation pulses, a clear transition from spontaneous emission to single-mode lasing is observed upon increasing the excitation level, with a typical threshold pulse energy of $P_{th} \sim 9$ pJ per pulse for lasing (Supplementary Fig. 1). To study the ultrafast emission and coherence properties of the NW lasers, we performed time-resolved pump–probe spectroscopy with non-resonant excitation as a function of the energy of the pump ($P_{pump}$) and probe ($P_{probe}$) pulses, respectively. Hereby, the NW lasers were excited using approximately 250-fs duration laser pulses at a repetition frequency of 80 MHz tuned to selectively excite the active GaAs core region of the NW non-resonantly at $\hbar \omega_{exc} = 1.59$ eV[7,8,10]. The temporal separation between pump and probe was precisely tunable over the range $\Delta t = \pm 100$ ps using an optical delay line that provides a relative precision better than 10 fs and the experimental observable was the time-integrated emission spectrum averaged over $> 10^6$ pump–probe excitation pulse pairs.

**Numerical modelling.** Our model is adapted from the well-known semiconductor Bloch equations for lasers[14,15] with an additional equation describing the number of carriers in the reservoir $N_2$ and the respective incoherent scattering processes from the reservoir to the lasing state[16]. We emphasize that phase coherence is not expected to be transferred from the excitation source to the NW laser output, since the pump pulse with an energy of $\hbar \omega_{exc} = 1.59$ eV is absorbed in the reservoir and subsequently incoherent carrier relaxation processes lead to the filling of the lasing state at $\hbar \omega_{em} = 1.51$ eV. The system of equations that are then solved read:

$$\dot{p} = -i\Delta\omega\, p - \frac{p}{T_P} - iE \frac{\mu_0}{2\hbar}(2\rho_1 - 1) \quad (1)$$

$$\dot{E} = -ig_m p - \kappa E + D\xi \quad (2)$$

$$\dot{\rho}_1 = R(\rho_0 - \rho_1) - \frac{\rho_1}{T_1} + Im\left(pE^* \frac{\mu_0}{\hbar}\right) \quad (3)$$

$$\dot{N}_2 = J - R(\rho_0 - \rho_1) - \frac{N_2}{T_2}, \quad (4)$$

where the complex electric field amplitude $E$ is given with respect to the rotating frame of the optical frequency $\hbar \omega = 1.5$ eV. The detuning $\Delta\omega$ gives the difference between $\hbar\omega$ and the lasing transition $\hbar\omega_{em} = 1.51$ eV, $p$ is the microscopic polarization of the lasing transition and $\rho_1$ the respective occupation probability. The number of carriers excited within this lasing level is given by $\rho_1 n^e$ with the total carrier density $n^e = 2.4 \cdot 10^{17}$ cm$^{-3}$ that can be accommodated by the lasing state of the NW. The dominating timescales are the photon lifetime $\tau_{ph} = (2\kappa)^{-1} = 1$ ps, the polarization lifetime $T_P = 5$ ps and the electron lifetimes of reservoir and lasing level, $T_2$ and $T_1$, respectively. Within the reservoir, we assume fast phonon scattering $T_2 = 10$ ps while within the lasing level the electrons are lost via spontaneous emission, that is, $T_1 = 0.6$ ns. The relaxation between the two electronic levels is implemented via a relaxation rate approximation, with a relaxation rate of $R^{-1} = 2.5$ ps to reach a quasi-equilibrium occupation of $\rho_0$ (given by a Fermi distribution) within the lasing level. The spontaneous emission factor ($\beta$) was chosen to be $\beta = 0.01$. For the numerical integration, the noise is implemented as a stochastic Gaussian white noise source $\xi$ with noise strength $D = \sqrt{\frac{\beta \rho_1 n^e}{T_1} \frac{\hbar\omega}{2\varepsilon_{bg}\varepsilon_0}} = \sqrt{\frac{\beta \rho_1}{T_1} g_m \frac{\hbar}{\mu_0}}$ refs [17,18]. The parameter $g_m$ is the coupling constant

that determines the gain of the light–matter interaction. It depends on the density of available electrons within the lasing transition $n^e$ and on the dipole moment $\mu_0 = 0.16\,\text{nm} \cdot e_0$ of the transition (both have been chosen to yield the gain observed in experiment). The carriers are injected into the reservoir with a pulsed pump intensity $J = J_{pump}e^{-\left(\frac{t-t_0}{\Delta_p}\right)^2} + J_{probe}e^{-\left(\frac{t-t_1}{\Delta_p}\right)^2}$ at times $t_0$ and $t_1$ with a pump–pulse width of $\Delta_p = 100\,\text{fs}$ according to the experimental setup. $J_{pump}$ and $J_{probe}$ are measured relative to the threshold pump rate to enable a comparison with the experiments.

The value chosen for $g_m$ yields a rate equation gain (modal gain) of $280\,\text{cm}^{-1}$ for a detuning of $\Delta\omega = 1\,\text{THz}$. To produce the data presented in the main manuscript, carriers are first injected into a reservoir and subsequently relax to the optically active level. The input parameters for gain and lifetimes (photons, electrons) were chosen according to experimental details: for example, for the data presented in Figs 1, 3 and 4 we chose a photon lifetime $\tau_p = 1\,\text{ps}$, a lifetime of electrons within the lasing level $T_1 = 0.6\,\text{ns}$, an electron lifetime inside the reservoir $T_2 = 10\,\text{ps}$ and a relaxation rate between reservoir and lasing level of $R^{-1} = 2.5\,\text{ps}$. The polarization lifetime of $T_p = 5\,\text{ps}$ was adjusted to reproduce the experimental results (longer/shorter $T_p$ lead to more/less visible Rabi oscillations).

**Data availability.** The authors declare that the data supporting the findings of this study are available within the article and its Supplementary Information files. And all data are available from the authors upon request.

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

## Acknowledgements

We acknowledge financial support from the DFG via the cluster of excellence Nanosystems Initiative Munich and project FI 947/4 and the DFB in the framework of SFB787.

## Author contributions

The project was conducted by J.J.F. and G.K. The theoretical interpretation and the numerical simulations were provided by K.L. and B.L. The experiments were performed by B.M. and S.S. The experimental setup was designed by M.K. and A.R. and was built by A.R., T.S., S.S. and B.M. The paper was written by B.M., J.J.F. and K.L. with contributions from all the authors.

## Additional information

**Competing interests:** The authors declare no competing financial interests.

**Publisher's note**: 

