## [Peer Review File · Nature Communications]

Reviewers' comments:

Reviewer #1 (Remarks to the Author):

The authors explore the coherent response of a single nanowire laser and reveal the emission of two mutually coherent laser pulse stimulated by pump and probe pulses respectively. They observe pulse separated by more than 30 ps, significantly longer than the pulse duration that they measure. This is a very nice result and will generate significant interest in the field.

They interpret the ability to generate mutually coherent pulses as being due to the coherence being stored in the Rabi oscillations that remain present after the initial lasing pulse. There are several aspects of this that are not clear and I have some questions.

How is coherence "preserved in the coupled electron-photon system"? The Rabi oscillations or "continuous absorption and stimulated emission" themselves cannot maintain the coherence of the system. Rabi oscillations suggest coherent driving of the electronic system, which leads to coupling between the photons and the electrons. The properties of the photon and electron may then be mixed to give values somewhere in-between the values for the pure electron and photon. The authors give a photon lifetime in the cavity of just 1ps, does this imply that the electron coherence time is long? or that the "amount" of coherence required is so small that there is still enough after 30 times the lifetime? At what temperature were these pump-probe measurements done? What sort of coherence properties are expected for the electrons? Why do the "Rabi oscillations" occur about the near transparency region and not go from zero to 100% excited state population? I could think of some possibilities, but this is not clear from the manuscript.

In the Fourier transform of the data presented in fig 3, there are several aspects that are troubling, based on the interpretation as given: there are significant differences between the data in the LL and L-SE plots in the region between the two pulses, that is, after the first pulse and before the second pulse. Indeed in the L-L case the signal changes as a function of Δt . This suggests that the probe pulse is affecting the emission before it has even arrived, which is clearly not reasonable. I would suggest that this is due to ambiguities in the Fourier transform.

As the authors state in the supplemental material, there is an ambiguity in the time ordering of pulses in the Fourier transform from the amplitude only information. However, the ambiguity actually goes even further than that, indeed there are actually an infinite number of solutions to a Fourier transform of real only data (for example, this is the major challenge in determining molecular structure from x-ray scattering). I would suggest that this may be the reason for the oscillations as a function of Δt at values of t between the two pulses. This may also raise questions about the quantitative details in fig 4a, as the phase of the oscillations appear to change when they shouldn't (as expected intuitively and from the simulations where this part of the signal didn't change as a function of delay).

Another question is regarding the arrival time of the second pulse in the L-SE case, which appears to give a slope with gradient less than 1, and less than that for the L-L case. Is this accurate? For a pump probe delay of 30ps this then gives a separation between lasing pulses of 20ps. Is this simply due to the time for onset of playing from the second pulse? Does this change as a function of delay and energy of the probe?

Whilst the complete interpretation of all these features seen in the experimental results may reasonably be beyonds the scope of this manuscript, the authors should at least acknowledge that the explanation they present is, at best, incomplete.

At what temperature were the pump probe experiments performed?

Reviewer #2 (Remarks to the Author):

This study found that there is a long-term mutual phase locking phenomenon by a GaAs-AlGaAs semiconductor nanowire laser. By a pump-probe method, the pulse pairs remain mutually coherent at a timescale about 30 ps, which is much longer than the emitted laser pulse duration time (3ps). Furthermore, the emitted laser pulse pairs of this phase-locked picosecond laser can achieve a repetition frequency up to ~ 200 GHz. Authors also interpreted the coherent phase information transformation between two laser pulses due to the coherent Rabi oscillations. This result shows the possibility to integrate a phase locking nanowire laser onto a common photonic circuit. This manuscript is written in good logical. I suggest accepting it after minor revision.

Comment 1:

The authors mentioned that the two-pulse interference actually have reported for GaN, CdS, ZnO NW-lasers in operation regimes, however the interference observed from the above nanowire is corresponding to the direct temporal overlap between the emitted laser pulses (lines 80-82 in page 3). In contrast to the results in this manuscript, the two-pulse interference phenomenon can exist even when pulse is separated by a longer time scale (one order of magnitude longer than the emitted laser pulses themselves). So, is this phenomenon can only be observed in this unique GaAs-AlGaAs nanowire? Is that possible to observe it in any other nanowire? What is the main factor to successfully observe this phenomenon, material or optical technique? I suggest authors give a brief description in appropriate location in the manuscript.

Comment 2: In the abstract, the laser pulse duration is ≤ 3 ps, however, this duration time is described as < 2 ps in summary (line189). So, which value is more accurate?

Comment 3: In line 79, "The data indicate that extremely high maximum repetition rates $\Delta f > 200$ GHz are possible, corresponding to emitted pulse durations $t_{\text{pulse}} < 3$ ps", it looks an inaccurate calculating from '3 ps' to '200GHz', because in line 143, a similar description is shown, where the calculating is from '5ps' to '200GHz'.

Comment 4: In line 122, there looks like a typo error of comma symbol.

Reviewer #1 (Remarks to the Author):

The authors explore the coherent response of a single nanowire laser and reveal the emission of two mutually coherent laser pulse stimulated by pump and probe pulses respectively. They observe pulse separated by more than 30 ps, significantly longer than the pulse duration that they measure. This is a very nice result and will generate significant interest in the field. They interpret the ability to generate mutually coherent pulses as being due to the coherence being stored in the Rabi oscillations that remain present after the initial lasing pulse. There are several aspects of this that are not clear and I have some questions.

1) How is coherence “preserved in the coupled electron-photon system”? The Rabi oscillations or “continuous absorption and stimulated emission” themselves cannot maintain the coherence of the system. Rabi oscillations suggest coherent driving of the electronic system, which leads to coupling between the photons and the electrons. The properties of the photon and electron may then be mixed to give values somewhere in-between the values for the pure electron and photon. The authors give a photon lifetime in the cavity of just 1ps, does this imply that the electron coherence time is long? or that the “amount” of coherence required is so small that there is still enough after 30 times the lifetime?

We thank the referee for this interesting and important question. As the referee mentions, the photon lifetime is very short (only ~1ps) but we observe preservation of coherence over much longer timescales. We describe here a simple (but intuitive) picture that helps to understand how these seemingly counterintuitive facts can be reconciled by considering the coupled electron-photon system. When the NW laser is operating far above threshold, the carrier density is high and a strong population inversion exists. In this regime, photons are rapidly generated in the cavity mode via stimulated processes, enhancing the coherence of the cavity field and the contribution from spontaneous (incoherent) processes are entirely negligible by comparison. The cavity field has a high degree of coherence due to the dominance of stimulated processes and high photon density in the lasing mode. As the excitation dies away, the gain cannot compensate for the losses in the system, lasing ceases the cavity field gradually loses coherence due to the phase fluctuations introduced by spontaneous processes. Importantly, reference to the blue curves shown in fig 4a and 4b of our manuscript shows that the inversion in the electronic system remains *very close to ~50% for long timescales* (up to ~30ps) and, thereby, **a large fraction of the light generated in the cavity is still produced by stimulated processes**, even though self-sustaining lasing is no longer possible since modal gain does not compensate for the losses in the system. If the second probe pulse is incident on the non-lasing NW at a point in time where *both* stimulated and spontaneous processes add photons to the cavity field, then the second pulse enhances the population

above inversion again (see fig 4b) and the phase of the lasing oscillation is again defined by the residual coherence in the cavity field due to the remaining stimulated processes. The experiments show that the timescales over which stimulated emission dominates over the spontaneous noise are for timescales $<30\text{ps}$, consistent with our experimental observations (figs 1 and 2) and theoretical modelling (data presented in fig 4, together with figs 1,3). To explicitly answer the referees question, we argue that the residual coherence in the coupled electron-cavity field is still sufficient to define the phase of the lasing oscillation after up to a few 10ps lifetimes. Some text pertaining to the above discussion was already present in the previous version of the manuscript (Methods Section) but we have attempted to make this clearer by substantially rewriting the text on p5-7 of the revised version of the manuscript and adding quantitative detail regarding our modelling into the main body of the manuscript. We hope that this will make this key point explicitly clear to all readers.

2) At what temperature were these pump-probe measurements done?

This information was indeed missing from the previous version of the manuscript and we thank the referee for pointing this out. We added the phrase “*All measurements reported in this paper were performed at $\sim 20\text{K}$.*” on p2 of the modified manuscript to explicitly point this out.

3) What sort of coherence properties are expected for the electrons?

The microscopic polarization of the optical transitions stores the optical phase information coherently. In the simulations reported in figs 1,2 and 4 the dephasing time of the polarization is chosen to be 5 ps – a typical value for both bulk GaAs and GaAs/AlGaAs QWs at low temperatures. Here, we added a comment and two references pertaining to bulk and 2D GaAs.

4) Why do the “Rabi oscillations” occur about the near transparency region and not go from zero to 100% excited state population? I could think of some possibilities, but this is not clear from the manuscript.

Rabi-oscillations in a coherently driven optical transition that start in the ground state would indeed lead to a complete inversion. In our case, the Rabi oscillations start in a partly inverted state after the laser pulse is emitted by the active medium itself (see figure 4 and the new discussion in the manuscript p5-7), which limits the oscillation amplitude of the occupation probability. Furthermore, the damping of the oscillations due to dephasing reduces their amplitude further, restricting the inversion to values close to transparency. The additional discussion added on p6-7 of the manuscript now makes this point explicitly clear.

5) In the Fourier transform of the data presented in fig 3, there are several aspects that are troubling, based on the interpretation as given: there are significant differences between the data in the LL and L-SE plots in the region between the two pulses, that is, after the first pulse and before the second pulse. Indeed in the L-L case the signal changes as a

function of Δt . This suggests that the probe pulse is affecting the emission before it has even arrived, which is clearly not reasonable. I would suggest that this is due to ambiguities in the Fourier transform. As the authors state in the supplemental material, there is an ambiguity in the time ordering of pulses in the Fourier transform from the amplitude only information. However, the ambiguity actually goes even further than that, indeed there are actually an infinite number of solutions to a Fourier transform of real only data (for example, this is the major challenge in determining molecular structure from x-ray scattering). I would suggest that this may be the reason for the oscillations as a function of Δt at values of t between the two pulses. This may also raise questions about the quantitative details in fig 4a, as the phase of the oscillations appear to change when they shouldn't (as expected intuitively and from the simulations where this part of the signal didn't change as a function of delay).

Here, we entirely agree with the referee that it is unreasonable that the probe pulse affects the emission before it has arrived! As such we purposefully did not interpret the detailed intensity variations of the interference fringes observed in fig 1, whereby the intensity of a particular interference fringe clearly exhibits beating that becomes stronger as the energy of the second pulse increases (not reproduced by our simulations), or fig 3 where the intensity of the oscillations between the two emission pulses exhibit oscillations as a function of Δt for a fixed $\Delta\tau$. The Δt dependent fine structure observed in this range for fixed $\Delta\tau$ is clearly aliasing arising from the discrete Fourier transform of the purely real data. This is confirmed by examining the fringes before the SE-L region of the data in Fig 3a ($\Delta t < 0$) for which the first weak (probe pulse) does not produce lasing and only one lasing pulse is generated by the second pulse. Here, the $\Delta\tau$ dependent oscillations are still observed but now with **no dependence on Δt** (i.e. no fine-structure). This observation strongly supports our identification of the oscillatory dependence as being due to damped Rabi oscillations and confirms that no systematic error occurs in our experiment. We added text to the manuscript to make this point explicitly clear.

We note that our methods of mapping from the frequency to the time domain are precisely the same as those used in ref [11], where the same analysis method was used to probe ultrafast gain dynamics in the emission from ZnO nanowires. Furthermore, the details referred to by the referee in fig. 4a are **not sensitive** to these phase ambiguities since the data presented is a one-pulse experiment. Ambiguities only occur if two laser pulses interfere and modulate the emission spectra. This is *not* the case for the single pulse experiment presented in fig.4a.

6) Another question is regarding the arrival time of the second pulse in the L-SE case, which appears to give a slope with gradient less than 1, and less than that for the L-L case. Is this accurate? For a pump probe delay of 30ps this then gives a separation between lasing pulses of 20ps. Is this simply due to the time for onset of playing from the second pulse? Does this change as a function of delay and energy of the probe?

Here, the referee raises a very interesting point that most likely reflects phenomena that go

significantly beyond the scope of the modelling we have performed. As mentioned above, one has to be careful making detailed interpretation based on the colormap representation presented in fig 3. In this respect, we also noticed a potential inconsistency in our definition of Δt in the previous version of the manuscript and have inverted $\Delta t \rightarrow -\Delta t$ such that $\Delta t = t_{pump} - t_{probe}$ in the sense of “pump-probe” (and not probe-pump) spectroscopy. Furthermore, the sign convention of Δt in Figs 1-3 is now consistent with Fig 4 to prevent the reader from being misled. To specifically address the question posed by the referee, we present below two curves recorded for $\Delta t=+25$ ps in the L-SE and L-L regimes that clearly show the first and second pulse emissions, together with the ambiguous oscillations between them.

Fig – Plotted amplitude of the $\Delta\tau$ dependence of the data shown in Fig 3 for $\Delta t=+25$ ps

The figure above shows a cross section of the data presented in Fig 3 at $\Delta t=+25$ ps, clearly showing the impact on the carrier dynamics of varying the intensity of the probe pulse. Clearly the onset of the second pulse is shifted by $\sim +10$ ps upon increasing the intensity of the probe pulse into the lasing regime. This may reflect the impact of the pulse powers on carrier relaxation and thermalisation that are not included in our modelling. However, we note that the maximum of the amplitude of the second pulse in the $\Delta\tau - \Delta t$ space clearly shows a linear dependence, as confirmed by the plot below.

Fig - Maximum intensity of the second pulse in the $\Delta\tau - \Delta t$ space plotted as a function of Δt , showing the linear variation of $\Delta\tau - \Delta t$ alluded to by the referee for both L-SE and L-L pump-probe regimes. The blue lines denote a linear variation.

We note that in the L-SE regime there is no visible deviation from linearity at larger Δt (>20 ps) indicating that carrier relaxation and thermalisation dynamics occur on a timescale of ca. 20ps, not at all unreasonable for GaAs (see new refs. 24-26 in manuscript). In the L-L regime, the probe pulse is sufficiently strong to reestablish lasing rapidly, even if relaxation and thermalisation is only partially complete and the $\Delta\tau - \Delta t$ curve is linear over the whole range plotted.

We speculate that any deviations from a unity gradient may be a sign that carrier relaxation from the reservoir to the lasing state is dependent on the population of the lasing state. This would be the case, e.g. in stimulated (parametric) scattering processes. Whilst the absence of an obvious parametric two-particle scattering process in the bulk band that conserves energy and momentum, the possibility for complex few-particle scattering processes certainly cannot be excluded. Furthermore, the relaxation efficiency from the reservoir to the “band” lasing states depend on final state occupation (Pauli blocking) and, whilst the final state carrier distribution is given by a Fermi distribution, the relaxation from the reservoir to the lasing state is non-Fermionic.

Whilst the complete interpretation of all these features seen in the experimental results may reasonably be beyonds the scope of this manuscript, the authors should at least acknowledge that the explanation they present is, at best, incomplete.

We agree and have modified the text pertaining to comparison of experimental data and our modelling (“excellent agreement” → “good agreement”, “quantitative → qualitative”). Furthermore, we have added a comment about the potential limitations of our modelling insofar as it does not include any reservoir saturation effects or power dependent time constants, dephasing times etc that would be necessary to achieve a good quantitative agreement. However, we note that the model certainly explains the reason for the long-term mutual phase locking of the emitted laser pulses over timescales much longer than the emitted pulse duration – the major focus and point of novelty of our manuscript.

At what temperature were the pump probe experiments performed?

As indicated above, all optical measurements reported in the paper were performed at a lattice temperature of $T \sim 20\text{K}$. A comment to this effect was added to the revised version of the manuscript.

Reviewer #2 (Remarks to the Author):

This study found that there is a long-term mutual phase locking phenomenon by a GaAs-AlGaAs semiconductor nanowire laser. By a pump-probe method, the pulse pairs remain mutually coherent at a timescale about 30 ps, which is much longer than the emitted laser pulse duration time (3ps). Furthermore, the emitted laser pulse pairs of this phase-locked picosecond laser can achieve a repetition frequency up to ~200GHz. Authors also interpreted the coherent phase information transformation between two laser pulses due to the coherent Rabi oscillations. This result shows the possibility to integrate a phase locking nanowire laser onto a common photonic circuit. This manuscript is written in good logical. I suggest accepting it after minor revision.

1)The authors mentioned that the two-pulse interference actually have reported for GaN, CdS, ZnO NW-lasers in operation regimes, however the interference observed from the above nanowire is corresponding to the direct temporal overlap between the emitted laser pulses (lines 80-82 in page 3). In contrast to the results in this manuscript, the two-pulse interference phenomenon can exist even when pulse is separated by a longer time scale (one order of magnitude longer than the emitted laser pulses themselves). So, is this phenomenon can only be observed in this unique GaAs-AlGaAs nanowire? Is that possible to observe it in any other nanowire? What is the main factor to successfully observe this phenomenon, material or optical technique? I suggest authors give a brief description in appropriate location in the manuscript.

Here, we have added a comment pertaining to the observation of 2-pulse interference in other nanowire systems when the emitted pulses exhibit temporal overlap in the detection system and, furthermore, have extended the discussion of the mechanism for the long-term phase locking. The main factor is that the laser remains close to transparency after lasing has ceased such that stimulated emission remains dominant and coherence can be maintained in the coupled electron-photon system. Other materials such as II-VI nanowires interact more strongly with light and, thereby, have a shorter spontaneous emission lifetime. Thus, this condition cannot be guaranteed and the phase coherence is preserved over shorter timescales.

Comment 2: In the abstract, the laser pulse duration is ≤ 3 ps, however, this duration time is described as < 2 ps in summary (line189). So, which value is more accurate?

Here we apologise for this inconsistency and have replaced the pulse duration by ~3ps everywhere in the manuscript.

Comment 3: In line 79, “The data indicate that extremely high maximum repetition rates $\Delta f > 200\text{GHz}$ are possible, corresponding to emitted pulse durations $t_{\text{pulse}} < 3\text{ ps}$ ”, it looks an inaccurate calculating from ‘3 ps’ to ‘200GHz’, because in line 143, a similar description is shown, where the calculating is from ‘5ps’ to ‘200GHz’.

Here, we note that the pulse width is not necessarily repetition rate since the pulse width and temporal profile depends on the intricacies of the electron-photon interaction in the pumped system (see fig 4). We have checked this and made the numbers fully consistent with the manuscript.

Comment 4: In line 122, there looks like a typo error of comma symbol.

Many thanks to the referee for noticing this ! – we have corrected it accordingly

REVIEWERS' COMMENTS:

Reviewer #1 (Remarks to the Author):

The authors have thoroughly addressed the questions raised in my previous review and the changes to the manuscript make the points regarding the mechanism for maintaining coherence substantially clearer. I now have no hesitation in recommending the manuscript be accepted and am sure it will generate much interest and further work to explain some of the details.

Reviewer #2 (Remarks to the Author):

The authors reported a long-term mutual phase locking phenomenon by a GaAs-AlGaAs semiconductor nanowire laser. By a pump-probe method, the pulse pairs remain mutually coherent at a timescale about 30 ps, which is much longer than the emitted laser pulse duration time (3ps). Furthermore, the emitted laser pulse pairs of this phase-locked picosecond laser can achieve a repetition frequency up to ~ 200 GHz. This result shows the possibility to integrate a phase locking nanowire laser onto a common photonic circuit. The authors have responded reasonably to all the comments and also have revised the manuscript. I suggest accepting it.